# A Rose by Any Other Name would *not* Smell as Sweet:
# Social Bias in Name Mistranslations

**Sandra Sandoval**
University of Maryland
sandracs@umd.edu

**Jieyu Zhao**
University of Southern California
jieyuz@usc.edu

**Marine Carpuat**
University of Maryland
marine@umd.edu

**Hal Daumé III**
University of Maryland
Microsoft Research
hal3@umd.edu

## Abstract

We ask the question: *Are there widespread disparities in machine translations of names across race/ethnicity, and gender?* We hypothesize that the translation quality of names and surrounding context will be lower for names associated with US racial and ethnic minorities due to these systems' tendencies to standardize language to predominant language patterns. We develop a dataset of names that are strongly demographically aligned and propose a translation evaluation procedure based on round-trip translation. We analyze the effect of name demographics on translation quality using generalized linear mixed effects models and find that the ability of translation systems to correctly translate female-associated names is significantly lower than male-associated names. This effect is particularly pronounced for female-associated names that are also associated with racial (Black) and ethnic (Hispanic) minorities. This disparity in translation quality between social groups for something as personal as someone's name has significant implications for people's professional, personal and cultural identities, self-worth and ease of communication. Our findings suggest that more MT research is needed to improve the translation of names and to provide high-quality service for users regardless of gender, race, and ethnicity.

## 1 Introduction

When people see their names incorrectly displayed by the technologies they use, this creates an unnecessary burden for them. Over time, the regular experience of an AI system getting a person's name wrong can have insidious detrimental effects such as the erosion of cultural identity and self-worth, similar to effects from racial microaggressions such as name mispronunciations experienced in the classroom (Kohli and Solórzano, 2012). Such experiences have the potential to be

| Source: | **Journee** es una poeta británica de fuerza, claridad y oficio honesto. |
|---|---|
| Translation: | **Journee** is a British poet of force, clarity and honest craft. |
| MT Output: | **Girls** are a British poet of strength, clarity and honest craft. |
| Error(s): | Name completely mistranslated |
| Source: | Приятно с вами познакомиться, **Амия**. |
| Translation: | Pleased to meet you **Amiyah**. |
| MT Output: | I'm happy to meet you **mom**. |
| Error(s): | Name translated as common noun |

Figure 1: Two hypothetical input sentences, originally in Spanish and Russian, together with correct English translations as well as the output of MT systems from our study; both mistranslate a person's given name.

frequent in online settings, whether via LinkedIn, professional email, Twitter, or Slack, or any other place where people address others or are addressed by name. In multilingual personal or workplace settings, displaying names incorrectly due to mistranslations as in Figure 1, whether in greetings or in the context of larger statements related to a person's name, can misrepresent a person's work, make professional communications more difficult, or erode their sense of identity and self-worth. This can lead to both allocational harms and representational harms (Crawford, 2017; Barocas et al., 2017), especially when rates of name mistranslation are spread disparately across social groups (Blodgett et al., 2020) (see section 2).

Unfortunately, the subtle but crucial detail of having correct name translations for everyone seems to have been overlooked. For instance, a prominent survey paper on gender bias in machine translation (MT) (Savoldi et al., 2021) presents a range of work that, while detailing underrepresentation of social groups in language, narrowly defines the problem as relating to misrepresentation of women with respect to linguistic expressions

about them (for example, incorrect pronoun usage) or the frequent mistranslation of female named entities into male entities. This reflects the tendency of research on gender bias in NLP to be focused almost exclusively on personal pronouns. A major exception is work in named entity recognition and coreference resolution, where various research has found that the accuracy of name recognition tends to deteriorate across gender as well as race and ethnicity (Mishra et al., 2020).

In this paper, we ask: *Are there disparities in the accuracy of machine translations of names across race, ethnicity, and gender?* We define and evaluate a particular form of bias through inequality in the machine translations of names. To achieve this, we constructed a dataset, Diverse Names in Context (DNIC), of English sentences, by combining templates with names that are known to be strongly associated with particular social groups (section 4). Using this dataset, we develop a procedure for analyzing the quality of machine translation systems' ability to correctly translate names and their contexts based on round-trip translation from and to English, through Spanish, Russian, Arabic and Chinese (section 5). We find that name translation errors across three state-of-the-art machine translations systems are inequitably impacting those with ethnic minority names, particularly Black (odds ratio=0.69) and Hispanic females (odds ratio=0.62), at a much higher rate than White males (section 6).

## 2 Sociolinguistic Background

Given names tend to be highly personal, and also reflective of a person's position in social groups, often with strong associations to a particular gender, a particular race or ethnicity, a social class, a religion, and more (Pharr, 1993). Beyond simply marking someone's identity, given names also signal and reinforce individuality and the social position of the person being referenced (Jeshion, 2009).

Due to the importance of given names, their regular mistranslation can potentially have substantial negative impacts on a person's life. Allocative harms (Crawford, 2017; Barocas et al., 2017) can arise either because 1) some people are experiencing worse system behavior despite paying the same for the use of a system, or 2) when professional communication (e.g., in job-seeking or recognition of their work) is made more difficult due to lower quality translations of their names and sentences written about them.

Name mistranslation also gives rise to significant representational harms, including, at least, harms related to quality of service, denial of self-identity, and erasure and alienation of both the name and its associated culture (Katzman et al., 2023).

Quality-of-service harms resulting from name mistranslation are clear—if a machine translation system works less well for one person's name, or names strongly associated with marginalized social groups, than others, those people and those groups experience a less effective machine translation system. This is particularly problematic when the user is less likely to be able to identify translation errors, such as when the target language uses a different alphabet than the source. This type of harm parallels those studied by Dyal-Chand (2021) in the context of incorrect auto-complete for some names over others, where names are corrected to more anglicized versions or translated to nouns. Dyal-Chand (2021) emphasizes that to mitigate the danger of reinforcement of structural racism through technologies, we must acknowledge the role that they have on all individuals stating that everyone has "... a right to full and equal use of these technologies".

Denial of self-identity (the inability to express oneself as desired), erasure (a system's routine removal of traces of some social groups), and alienation (a system marking some social groups as other) arise due to the close ties between one's name and one's culture and community. Kohli and Solórzano (2012) emphasize how racial microaggressions, such as teachers' mispronunciation or anglicization of of names, often leave people of color feeling diminished, feeling like they or their culture aren't valued, and presure them to take responsiblity to make it easier for others to pronounce their names. When viewed through the lens of AI systems, this pressure leads to a form of "language standardization" in technology (Pérez-Quiñones and Salas, 2021), a form of language ideology that can be used to justify social hierarchies (e.g., Alim et al., 2016; Rosa and Flores, 2017).

Because names can code for a multiplicity of social axes (gender, race, ethnicity, religion, social class, birth generation, etc.), it is important to study the impact of name mistranslation not just on a single social axis, but also based on how axes intersect. Intersectionality is a core concept in Black feminism, introduced in the Combahee River Collective Statements 1977; 1983: "Because the intersectional experience is greater than the sum of racism and

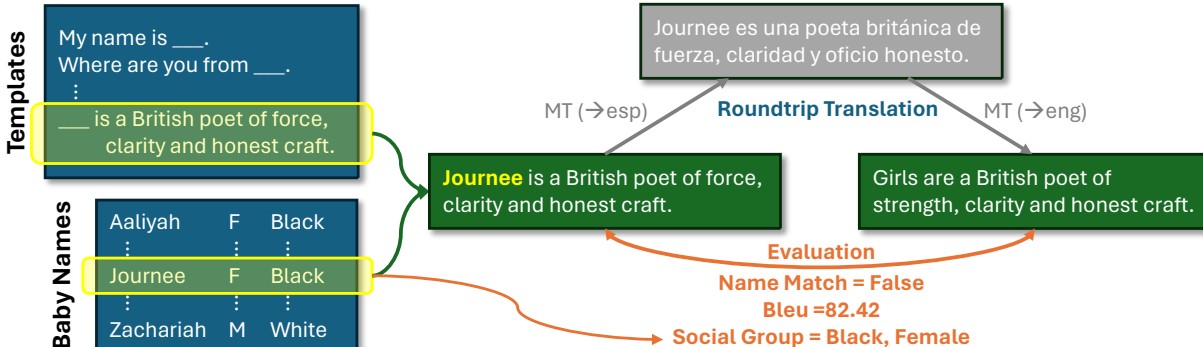

Figure 2: Evaluation approach: A template is combined with a baby name (with an associated social group) to produce an English sentence. This sentence is round-trip translated through a pivot language (in this case, Spanish) back into English. We measure (a) whether the exact input name (Journee) appears verbatim in the output and (b) the BLEU score, which get stored with the associated social group.

sexism, any analysis that does not take intersectionality into account cannot sufficiently address the particular manner in which Black women are subordinated." Intersectionality was applied in a legal setting by Crenshaw (1989) to analyze the ways in which U.S. antidiscrimination law fails Black women. In our context, we analyze name mistranslation across a combined race/ethnicity axis (White non-Hispanic, Black non-Hispanic, Hispanic, and Asian or Pacific Islander), and a binary gender axis (Male, Female), with the concomitant limitations arising from that coarse categorization.

## 3 Approach Overview

In Figure 2, we provide a visual depiction of our process flow for an example English sentence (template plus name) in the context of our contributions. Our goal was to assess if names get mistranslated at different rates across race/ethnicity and gender groups in realistic contexts. Thus, our steps were: 1) *Names*: We collected a list of names strongly associated with race/ethnicity and gender; 2) *Instantiated English Sentences (Names plus Templates)*: We then situated the names in sentence contexts (i.e., templates) that we thought would be found in the "real world"; 3) *Round-trip Translations (RT)*: Then we translated the complete English sentences (template + name) into four languages and back to English. We used RT translation to see if the final name in the translation output would be the same as the original name, as well whether the template portion remained in tact; Finally, 4) *Translation Evaluation Approach*: with our resulting RT translations, we could ascertain the impact of the combinations of race/ethnicity and gender of the names on the translations by assessing the strength and

significance of our measures (of the correctness of name translations, and of sentence translation quality as a whole). We used mixed effects statistical analysis to quantify these effects.

We describe each of these components in the context of the dataset we developed next.

## 4 Diverse Names in Context Dataset

To study the effect of mistranslating names on women and ethnic minority populations in the United States, we construct the DNIC dataset by collecting a diverse set of names and English sentence templates where name translation particularly matters. Instantiating these templates with different names let us isolate the effect of changing *just* the name. This builds on many existing datasets that seek to measure the potential harm caused by NLP systems across different social groups using templates instantiated with different names, pronouns, or other variables (Levesque et al., 2011; Garg et al., 2018; Cho et al., 2019; Rudinger et al., 2018).

### 4.1 Name Selection

We select names from birth records that are strongly associated with race/ethnicity and gender, as measured by Weight of Evidence (Good, 1985).

We started with a baby names list from the New York City Open Data "Popular Baby Names" website, which contained a record for all baby names by race/ethnicity and gender for births in 2019.[1] This list gave us a total of $1,935$ first name records where each had the following attributes (fields): the gender of the baby from birth records (Female

---

[1] https://data.cityofnewyork.us/Health/Popular -Baby-Names/25th-nujf

or Male), the birth parent's[2] race/ethnicity (White, Black Non-Hispanic, Asian and Pacific Islander, Hispanic), and the number of babies for that name, gender and race/ethnicity combination.

Since the same name can be given to babies of different gender and race/ethnicity, we determine how strongly associated each name is with each social group by computing its *weight of evidence* (WoE) in favor of a race/ethnicity and gender group. Good (1985) defines the WoE as how strong the evidence is in favor of a hypothesis. In our setting, the evidence is the name, and the hypothesis is the combination of race/ethnicity and gender and we ask how much evidence the name provides for social group (race/ethnicity and gender combination):

$$WoE(\texttt{group} : \texttt{name}) \triangleq \log \left[ \frac{P(\texttt{name} \mid \texttt{group})}{P(\texttt{name} \mid \neg\texttt{group})} \right]$$

This log-odds interpretation of the weight of evidence (Alvarez-Melis et al., 2021) is positive when the probability of the name evidence is higher under a specific group than under all other groups. Following Good (1985), we associate any name with a social group whenever $WoE(\texttt{group} : \texttt{name}) \geq 2$. Three example names with highest *WoE* are shown for each social group in Table 1 and overall statistics for the baby names dataset (including our *WoE*-filtered version) are given in Table 2[3].

## 4.2 Template Selection

Each template consists of an English sentence with a placeholder to be replaced by a first name. We select 16 templates (Table 3) where any of the names in our list can be inserted (i.e., lexical or syntactic gender does not code for a specific name), and which represent contexts where names and their mistranslations matter: 1) everyday interpersonal interactions, 2) professional biographies.

For personal interactions, we use 5 simple sentences from an English-Italian tourist phrasebook (DK, 2017) as well as one manually created sentence as another simple source of comparison. In these contexts, mistranslating a name has the potential to negatively impact communication, as well

| Social Group | | | |
|---|---|---|---|
| **Gender** | **Race/Ethnicity** | **Name** | **WoE**(g : n) |
| Female | AAPI | Inaaya | 17.77 |
| | | Tenzin | 17.77 |
| | | Joanna | 17.73 |
| | Black | Fatoumata | 18.21 |
| | | Wynter | 17.87 |
| | | Reign | 17.83 |
| | Hispanic | Alaia | 18.09 |
| | | Valerie | 17.40 |
| | | Alexa | 17.36 |
| | White | Chaya | 18.31 |
| | | Miriam | 18.23 |
| | | Rivka | 17.89 |
| Male | AAPI | Ayaan | 18.64 |
| | | Mohammad | 18.44 |
| | | Arham | 18.33 |
| | Black | Nasir | 18.29 |
| | | Malachi | 17.95 |
| | | Kenzo | 17.91 |
| | Hispanic | Angel | 18.31 |
| | | Gael | 18.27 |
| | | Thiago | 18.02 |
| | White | Moshe | 18.41 |
| | | Chaim | 17.89 |
| | | Mordechai | 17.73 |

Table 1: A sample of the names with the highest WoE for each race/ethnicity and gender group.

| | # Babies | | # Unique Names | | # Final Names | |
|---|---|---|---|---|---|---|
| | **F** | **M** | **F** | **M** | **F** | **M** |
| AAPI | 3370 | 4695 | 157 | 175 | 45 | 57 |
| Black | 3246 | 4418 | 172 | 168 | 94 | 66 |
| Hispanic | 7832 | 10566 | 254 | 264 | 113 | 117 |
| White | 12372 | 14197 | 383 | 362 | 231 | 211 |

Table 2: High-level statistics for the original baby names data as well as with Weight-of-Evidence filtering for values of two and greater ($WoE \geq 2$).

as the personal identity and sense of importance of the named participants.

We use English sentences from the professional biographies data set associated with (De-Arteaga et al., 2019). In these contexts, mistranslating a name can negatively impact professional communication, as well as the named person's recognition and lead to socioeconomic harms.

Finally, the DNIC dataset is constructed by instantiating each of the 16 templates with the 1,935 name records selected, yielding a total of 179,328 English sentences containing a name in context, where each name is strongly associated with a combination of gender and race/ethnicity.

---

[2]The original names data terms this the "Mother's ethnicity"; we assume that their use of "Mother" is equivalent to the term "birth parent" and reference it as such.

[3]The number of babies in the original names list data was thresholded such that only those with a count of 10 or greater (for name,gender and race/ethnicity combination) were included, which is why there appear to be far more male than female babies (apparently female babies have rarer names).

| # | Template |
|---|----------|
| 1 | They never go anywhere without ___. |
| 2 | Pleased to meet you ___. |
| 3 | My name is ___. |
| 4 | Where are you from ___. |
| 5 | What would you like to do ___. |
| 6 | ___ is an Associate Professor of Development Studies at Nelson Mandela University in South Africa. |
| 7 | ___ is a reputable dentist with over 20 years of experience helping patients of all ages care for their smiles through top-quality treatment. |
| 8 | ___ is a Software Engineer by passion and profession. |
| 9 | ___ is an underground magician whose technical sleight of hand prowess is widely-respected by magicians in the inner circle. |
| 10 | ___ is an Orthopedic Surgeon (Hand Surgeon) in Laredo, TX. |
| 11 | ___ is a painter who approaches the medium as a formal exercise. |
| 12 | ___ is a British poet of force, clarity and honest craft. |
| 13 | ___ is a Jungian psychotherapist with nearly 30 years of clinical experience. |
| 14 | ___ is an interior designer who has specialized in historic restorations and art consultation. |
| 15 | ___ is a behavioral nutritionist spcializing in the low-carb and keto lifestyle. |
| 16 | ___ is a Registered Dietitian with a graduate degree in nutrition and wellness. |

Table 3: The 16 templates chosen through manual review to have a selection with varying lengths and language usage. Phrase 1 was manually created, phrases 2–5 were from the tourist phrase book, and phrases 6–16 came from the bios dataset.

# 5 Methodology

We now describe how we use MT to translate the DNIC dataset (Section 5.1), and introduce our approach to automatically evaluate the resulting name translations (Section 5.2) and conduct a statistical analysis of the impact of social groups on name translation quality (Section 5.3).

## 5.1 Machine Translation Settings

To get a representative sample of real-world MT quality, we use three different MT systems: two widely used online translation services (Google Translate[4] and Microsoft Translator[5]) and the OPUS MT open models[6] based on Marian MT[7].

We translate the English sentences from the DNIC dataset into four diverse languages: Arabic, Chinese, Russian and Spanish. These languages

[4]https://cloud.google.com/translate/docs/reference/rest
[5]https://www.microsoft.com/en-us/translator/business/translator-api/
[6]https://github.com/Helsinki-NLP/Opus-MT
[7]https://github.com/marian-nmt/marian

were selected to represent diffferent language families, writing systems and typology. They are all high resource languages, for which we expect MT quality to be good enough on average to be useful, as indicated by BLEU scores on the FLORES benchmark (Figure 4).

## 5.2 Evaluating Name Translation Quality

Evaluating the quality of name translation raises some specific challenges in addition to all the difficulties that come with MT evaluation in general settings. Different people might have different criteria for defining what constitutes an acceptable translation of their name. Some might use the exact same name in the two languages. Others might want to see specific diacritics in Spanish but not in English, might expect their Chinese name to use specific characters, or might even use an entirely different first name in the target language. As a result, name translation quality cannot be directly evaluated by checking whether MT outputs match reference translations written by a third party. Instead, we propose a round trip translation approach to estimate name translation quality automatically, without collecting first person judgments.

**Round Trip Translation.** We evaluate the quality of each MT output on the DNIC dataset by translating it back into English using the exact same system used for the forward translation pass, and by comparing the resulting Round Trip (RT) output with the original input using different metrics.

While RT translation is rightly considered not to be a reliable approach to evaluate translation quality in the general case (Somers, 2005), it is well suited to our specific use case for several reasons. First, although roundtrip translation does not enable us to detect in which direction an error was introduced, any error that shows up in the round trip must have been caused by (at least) one error in one direction or the other. Thus, errors detected through RT translation represent a *lower bound* on the actual error rate. On the other hand, if a name comes back correctly in the round trip, it could be that it was correctly translated in both directions, or it could be that two errors conspired to lead to no error. So, while we may not pinpoint where any translation errors may occur with the roundtrip evaluation approach, we are able to deduce the minimum error rate of a name.

Finally, the computation of translation accuracy and quality in the final translation output relative

to the template would be straightforward and reasonably accurate, not requiring knowledge of non-English languages nor human review of each translation. In essence, round-trip evaluation allowed us to provide the closest approximation of an error rate possible for name translations using MT. Manual evaluation of name translations in each direction, on the other hand, would require a significant amount of expertise and labor.

**Name Translation Accuracy.** We count a name as translated correctly if and only if the RT translation contains a token that is a case-sensitive match to the original name from the names list.

**BLEU** To capture the impact of name mistranslations on the rest of the sentence, we computed Sacrebleu's (Post, 2018) sentence-BLEU using default settings[8] to score the back-translations against the original English sentences.

### 5.3 Measuring the Impact of Social Groups with Mixed Effects Models

We aim to estimate the effect of the social group (gender and race/ethnicity) associated with a name on our relevant outcome variables—name accuracy and sentence-BLEU—while controlling for all other effects: MT system, pivot language, and template. We adopt a parametric approach using generalized linear mixed effects models for this analysis, where we model the outcome as a linear combination of variables passed through a link function $f$. We represent each input variable as a binary indicator of gender (female or not), race/ethnicity (AAPI or not, Black or not, Hispanic or not), MT system, pivot language, and template id as[9]:

$$
y = f\Big(b + \underbrace{gG}_{\text{Baby Gender}} + \underbrace{z_2Z_2 + z_3Z_3 + z_4Z_4}_{\text{Birth Parent's Race/Ethnicity}} \quad (1)
$$
$$
+ \underbrace{d_1G{\times}Z_2 + d_2G{\times}Z_3 + d_3G{\times}Z_4}_{\text{Interactions between Gender and Race/Ethnicity}}
$$
$$
+ \underbrace{s_2S_2 + s_3S_3}_{\text{MT Systems}} + \underbrace{l_2L_2 + l_3L_3 + l_4L_4}_{\text{Pivot Languages}}
$$
$$
+ \underbrace{t_2T_2 + \cdots + t_{16}T_{16}}_{\text{Template Id}}\Big)
$$

---

[8]except for "effective order =True" which was necessary for sentence-BLEU calculations versus corpus-BLEU

[9]To avoid a redundant encoding, our reference (or "base") variables were: Male for Gender, White Non-Hispanic for race/ethnicity, Google for MT system, Spanish for language, and template 1 for template id. Effects are relative to these.

| Coefficient | | Odds Ratio | $\beta$ | p |
|---|---|---|---|---|
| $b$ | (Intercept) | 3.92 | 1.37 | 0.012* |
| $g$ | IsFemale | 0.92 | -0.08 | 0.000* |
| $z_2$ | IsAAPI | 1.20 | 0.18 | 0.000* |
| $z_3$ | IsBlack | 0.93 | -0.07 | 0.004* |
| $z_4$ | IsHisp | 1.21 | 0.19 | 0.000* |
| $d_1$ | IsFemale $\wedge$ IsAAPI | 0.87 | -0.13 | 0.001* |
| $d_2$ | IsFemale $\wedge$ IsBlack | 0.69 | -0.37 | 0.000* |
| $d_3$ | IsFemale $\wedge$ IsHisp | 0.62 | -0.47 | 0.000* |

Table 4: Results of Logistic Mixed Effects Regression with Name-Exact-Match Outcome Variable. Fixed effects shown. We also controlled for the random effects of template id, translation language, and MT system.

Here, capitalized letters indicate the indicator variables and lowercase variables represent estimated parameters. $b$ is a fixed intercept, and the outcome $y$ is either name exact match (in which case the link function $f$ is the logistic function) or sentence-BLEU (for which $f$ is the identity function).

In our analysis, we treat the social group variables (gender $g$, race/ethnicity $z_i$ and interactions $d_i$) as *fixed effects* (colored purple) and the incidental variables (MT system $s_i$, pivot language $l_i$ and template id $t_i$) as *random effects* (colored green). This choice is based on the definition for fixed and random effects by Searle, Casella, and McCulloch that "effects are fixed if they are interesting in themselves or random if there is interest in the underlying population" (Searle et al., 1992). In our setting, we were most interested in the effects of *gender* and *race/ethnicity* on our outcome variables and therefore consider these our fixed effects. We considered all other variables random effects since we expected them to randomly impact our translation quality and accuracy and we wanted to control for their effects in our model.

## 6 Results

To answer our primary research question, *"Are there widespread disparities in machine translations of names across race/ethnicity, and gender?"*, we analyzed our results to determine if names are mistranslated at higher rates for certain groups, and investigated any accompanying translation quality degradation of the entire sentence.

| Full Effect | Odds Ratio | $\beta$ | % Diff |
|---|---|---|---|
| Black Female | 0.59 | -0.53 | -40.91 |
| AAPI Female | 0.97 | -0.03 | -3.14 |
| Hispanic Female | 0.69 | -0.36 | -30.51 |

Table 5: The *full effect* takes into account the combined effect of each contributing coefficient for a race/ethnicity and gender combination and is a case-sensitive match. *% Difference* is calculated relative to White males (Odds = 1), the reference gender and race/ethnicity group.

| Coefficient | | $\beta$ | p |
|---|---|---|---|
| $b$ | (Intercept) | 22.81 | 0.467 |
| $g$ | IsFemale | 0.10 | 0.469 |
| $z_2$ | IsAAPI | 0.25 | 0.225 |
| $z_3$ | IsBlack | -0.25 | 0.204 |
| $z_4$ | IsHisp | 0.32 | 0.047* |
| $d_1$ | IsFemale $\wedge$ IsAAPI | -0.22 | 0.464 |
| $d_2$ | IsFemale $\wedge$ IsBlack | -0.83 | 0.001* |
| $d_3$ | IsFemale $\wedge$ IsHisp | -0.77 | 0.001* |

Table 6: Results of Linear Mixed Effects Regression with Sentence-BLEU Outcome Variable. We show the coefficients on the fixed effects including gender, race/ethnicity, and the interactions between them. We also controlled for the random effects of template id, translation language, and MT system.

## 6.1 Correctness of Name Translation

Table 4 shows the estimated odds ratios (and corresponding coefficients and p-values) for each of the fixed effects in the logistic mixed regression analysis of name translation accuracy. Here, we see that female-associated names ($g$) had significant negative effects ($Oddsratio = 0.92, \beta = -0.08$, $p = 0.00$) on the odds of having a correct name translation in the round-trip translation; in terms of odds ratio, the odds of a female-associated name mistranslation is eight percent greater than that of male-associated names. Furthermore, the odds of an AAPI-associated name being translated correctly is 20% higher than baseline ($OddsRatio = 1.2, \beta = 0.18, p = 0.00$).

The largest effects we see are the intersectional effects, which are exceptionally pronounced for Black and female-associated names and Hispanic and female-associated names. However, these have to be adjusted for the fact that, for instance, the effect of an AAPI- and female-associated name will be the sum of $g$ (IsFemale), $z_1$ (IsAAPI), and

$l_1$ (IsFemale $\wedge$ IsAAPI). This adjustment is shown in Table 5. Here, we see that Black and Female-associated names will be mistranslated about 41% more frequently than baseline, 31% percent more for Hispanic+Female-associated names, and 3% more for AAPI+female-associated names.

## 6.2 Overall Translation Quality

Beyond the (mis)translation of the name alone, we wanted to see how overall sentence translation quality varied for the dataset records corresponding to each social group, based on the name and its combination with the name attributes of gender and race/ethnicity (as measured by sentence-BLEU). We show some example sentences in Appendix Table 7 of degraded translation quality for sentences with Black female-associated names. The results of the linear mixed effects regression are shown in Table 6. Here, we see similar patterns as in the name translation accuracy, but with smaller effect sizes (as expected, since BLEU is an average over many words, mostly correctly translated).

We see significant ($p < 0.05$) effects for Hispanic-associated names (BLEU *increase* of 0.32, $p = 0.047$), as well as for Female+Black-associated names (BLEU *decrease* of 0.83, $p = 0.001$) and for Female+Hispanic-associated names (BLEU *decrease* of 0.77, $p = 0.001$). Similar to the adjustment in the case of name translation, the overall effect for Female and Black-associated names is $-1.08$ BLEU points, and for Female and Hispanic-associated names is $-0.45$ BLEU points.

## 6.3 Average BLEU and Name Translation Accuracy Across Systems

In our analysis we consider the machine translation system (as well as the pivot language and the template) to be random effects, and therefore do not present results related to the quality of each machine translation system on our task. Nonetheless, it is of potential interest how well each of these systems performs on our data. In terms of name translation accuracy, Google Translate has an average accuracy of 73%, Marian has 70%, and Microsoft Translator has 78%. In terms of BLEU, Google Translate has a score of 55.92, Marian has 40.72, and Microsoft Translator has 55.20 (compare to 41.01, 33.28, and 37.71 respectively on the FLORES benchmark in Appendix Figure 4).

## 6.4 Qualitative Insights related to Gender and Ethnicity in Roundtrip Translations

Based on the results above, we investigate what types of errors there are at a more fine-grained level than just names translated correctly versus not. To perform this analysis, we manually coded a randomly chosen sample of 300 names, split between our different social groups. Two were chosen to be "hard" for machine translation systems based on the previous results—Black Female- and Hispanic Female-associated names—and two were chosen to be "easy'—Hispanic Male- and AAPI-associated names. In Figure 3, we show estimates of the prevalence of each type of mistranslation for Black Female and Hispanic Female-associated names (the name types that were more difficult to translate) with rate fluctuation ranges based on a 95% confidence level. We do not observe a systematic difference between the "hard" and "easy" to translate social groups in terms of the distribution of types of errors, thus we show only the former.

A plurarility of the errors is *variant of name* – cases when the name was translated to slightly differently spelled variant of, arguably, the same name (e.g., Hanna/Hannah, Mohammed/Mohamad, Amirah/Amira). For these, either only a few characters were different, or it was labeled as such based on our cultural knowledge of name variants[10].

The next most common type of error is names that were translated as other names with some sounds in common, but which we did not judge to be a common name variation (e.g., Kamiyah/Camia, Brielle/Preel). These likely occur largely because of transliteration into non-Latin scripts with different phonetics.

A smaller category of errors includes words that are translated into common nouns, which often has small spillover effects into the translation of the rest of the sentence (e.g., "What would you like to

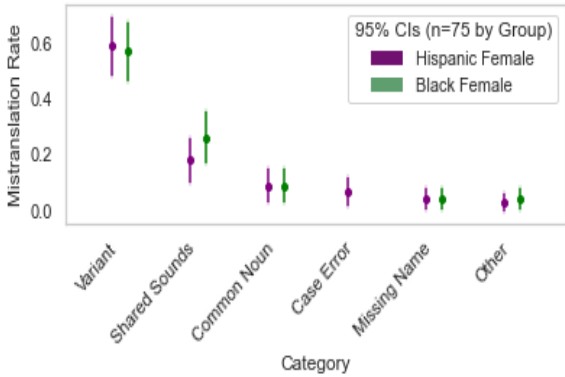

Figure 3: The *mistranslation rate* is a rough estimate based on the number of translation errors manually coded as a given category (x-axis) divided by the total number of randomly chosen mistranslated names for the social group (only the Black Female and Hispanic Female shown here since other categories had similar distributions of error types).

do Prince" → "What princes do you want to be", and "Miracle is a software engineer…" → "The miracle is a software engineer…').

A similar percentage of errors are case-errors (but not common nouns). This is a group in which the name is missing entirely (e.g., "My name is Eve" → "I'm summer night"). Occasionally the output contains non-Latin characters, again, likely due to a pivot through a language which uses a non-Latin script.

## 7 Conclusion

We explored how race/ethnicity and gender impact both the odds of obtaining a correctly translated name as well as the translation quality of a sentence. To accomplish this, we introduced a roundtrip-translation-based approach to evaluate the machine translation of names in context, as well as a dataset of Diverse Names in Context based on names highly associated with birth-derived gender and birth parent race/ethnicity.

Our results showed that female-associated names are likely to be mistranslated at significantly higher rates than male-associated names. Our intersectional analysis further demonstrates that Black Female- and Hispanic Female-associated names are mistranslated 30–40% more frequently than White Male-associated names. Furthermore, Black and Hispanic Female-associated names are associated with significantly worse overall translation quality as measured by BLEU.

These results hold, even when controlling for

---

[10] In our results, the most common type of error is a name variant, which could potentially be perceived as unimportant. There is substantial evidence that even small name variations (including misspellings) can cause harm. Freedberg (2002) observes "If names don't have the accents or the tildes, they are not spelled correctly". Lieberson and Bell (1992) note that "choosing a name is an inherently social process" influenced by a wide range of personal considerations, and "(name) choices are increasingly free to be driven by factors of taste...the choices will also reflect differences in taste between subsets of the population". Spelling of names is also often very closely tied to social groups; for example, Campbell (2023) observes that "Given names used by Black people are often invented or creatively-spelled variants of more traditional names." Overall, these translation errors are harmful and socially contingent and important to correct.

the template, MT system, and pivot language.

We have not performed a root cause analysis to determine from where the name translation disparities we observed arise, but expect that they may arise in part due to names of certain social groups being underrepresented in the training data upon which machine translation and large language models are trained (Weidinger et al., 2021). In addition, proper names, cross-linguistically, may have (for instance) inflectional forms by gender, or derivative forms originating from verbs; the way these forms are reflected in the training data may also influence mistranslation.

This work highlights the need to mitigate the bias that vulnerable social groups experience both personally and economically by seeing their names incorrectly displayed in translations. Despite the potential for harm arising from mistranslating names, as well as the fact that names are often translated differently from other word types, there has been relatively little work in machine translation that focuses specifically on names. (Maurel et al., 2007) is one example, which focuses specifically on the automated translation of *rare* proper names. Recognizing that some names should be translated and some should be transliterated, (Hermjakob et al., 2008) build a classifier to predict which is which, and to apply a different model for the two types.

We hope that this work highlights the danger that name mistranslations could have resulting in long-term harms (section 2). Given our findings, and in light of historical evidence that language technology has exacerbated harms for vulnerable populations (e.g., Weidinger et al., 2021; Bender et al., 2021; Blodgett et al., 2020; Savoldi et al., 2021), we suggest that lower translation accuracy for names of people from vulnerable social groups risks their experiencing both allocative harms (Crawford, 2017; Barocas et al., 2017)—such as poorer quality of MT system service or poorer professional communications when MT systems incorrectly translate their names (or sentences including their names)—and representational harms such as erasure of the name and its associated culture (Katzman et al., 2023).

## Limitations

Our analysis is limited by the nature of the baby names data we used: birth parent's ethnicity is not the same as child's ethnicity, and gender assigned at birth is not the same as the child's gender.

The gender categories (only male/female) and race/ethnicity categories are significantly limiting. Furthermore, by focusing on names of babies *born* in the United States, in addition to a US-centric analysis, the analysis here erases the experience of immigrants. We expect that if anything name translation errors are higher for immigrant's names (unless they choose an American name) due to the social trends of anglicization of names in the US – per Pinsker (2019), "in general, the names immigrants give their children go through three stages: from names in the original language to universal names, and finally to names in the destination-country language."

The second major limitation is that both evaluation measures we have are at best proxies for real harms incurred by people. Name translation error and BLEU score capture important *intrinsic* properties of the translation quality, but do not directly speak to the allocational or representation harms suffered as a result of those mistranslations.

Finally, it is worth noting that the templates do not fully represent the full variety of sentences that appear in real-world contexts. This is a standard limitation of template-based datasets. We chose our sentence templates in light of our harms, and specifically to align to scenarios where names matter and to be independent from the demographic variables considered, including ensuring gender neutrality of the sentences. Our statistical analysis through a mixed-effects model, where the template is treated as a random variable, also reflects the fact that this is a non-exhaustive, non-random set of possible templates.

## Ethics Statement

In this paper, we emphasize the importance of ethical machine translation that considers intersectionality, i.e., social groups defined but not limited by the characteristics of gender and race/ethnicity. We highlight the fact that MT technologies may reinforce structural racism if as a community we do not acknowledge and attempt to mitigate harms such as lesser quality of service, denial of self identity, and erasure of a name and its associated culture. We find that the use of state-of-the-art MT tools as seen in the mistranslations of names and their contexts inequitably impacts vulnerable social groups.

With respect to data considerations, we note that our first (given) names list was made public by the City of New York and therefore is open access, as

are the biographies we utilized for our templates; however, the tourist phrases were sourced from tourist phrase book but do not contain sensitive information. Our names list is a reasonably diverse representation of the population of New York City, but is not representive of the U.S. nor world population as a whole. Finally, for a name to be included in the names list, it must have been both a first name as well as been given to ten or more babies for that name, race/ethnicity and gender combination. Therefore, there is little risk of revealing personally identifiable information.

## Acknowledgments

First, this work was supported in part by the NSF Fairness in AI Grant 2147292, by the Google Award for Inclusion Research Program, and an NSF Computing Innovation Fellowship (latter for Jieyu Zhao). We thank these organizations for making work such as ours in fairness/bias research possible.

We greatly appreciate the time and care our reviewers, program chairs and areas chairs put into reading our paper and providing constructive feedback. In addition, we would like to thank Dr. Peter Rankel for his helpful feedback in the early stages of our development of our methodology, as well as the University of Maryland's Computational Linguistics and Information Processing Lab (CLIP) for feedback on our paper draft.

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

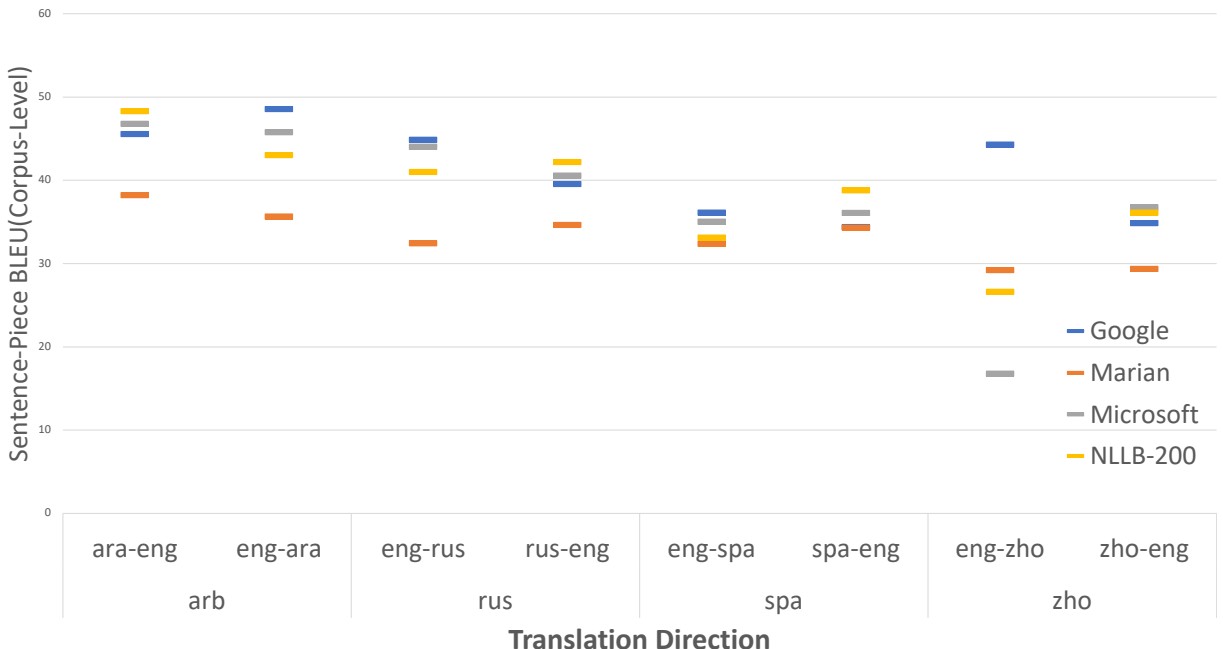

Figure 4: Translation quality of MT systems considered on the the FLORES 200 benchmark, as measured by Sacrebleu's sentence-piece corpus-level BLEU.

## A  Benchmarking MT Systems on the FLORES 200 Reference Translations

We ran Sacrebleu's sentence-piece corpus-level BLEU against Meta AI's FLORES 200 human-translated benchmark dataset (Figure 4) to gain insights into the variation of sentence-piece BLEU scores by NMT system and language (Costa-jussà et al., 2022). Similar to the benchmark sentence-piece BLEU scores, with our data we generally saw that Google outperformed Microsoft and Marian, with Marian machine translation having the lowest sentence-BLEU scores. The ranges for the benchmark sentence-piece BLEU scores were about 33 for Marian to 41 for Google, whereas our sentence-BLEU scores ranged from about 41 for Marian to 56 for Google.

## B  Why Mixed Effects Models: Additional Explanation

We utilized mixed effects models for a couple of reasons. Most critically, we believed that the variation in sentence-BLEU and name translation accuracy would be distinct for different groups (ex. for White females versus Black males) and explained by both fixed and random effects variables. In addition, we could not be sure that our observations were independent; in particular, we had multiple observations for each template, our unit of analysis. For example, "Camila is a painter who approaches the medium as a formal exercise" was the template for multiple observations where the observations differed in that they each had a unique combination of translation language, NMT system, and template. Therefore, the different values of sentence-BLEU and the "name exact match" outcome measures for each template could be correlated.[11]

## C  Model specification

Both models shared the same list of fixed effects and random effects, with the fixed effects being race/ethnicity and gender and the random effects being the MT system, language, and template. Our analysis dataset for input to our models consisted of the outcome variables (the evaluation metrics) and the indicators for each of our predictors, where each was a category of the variable (dimension). For example, the gender variable was 1 for a name with a female attribute and 0 for a name with a male attribute.

---

[11] https://github.com/kshedden/Statsmodels-MixedLM

| Input Sentence | Round-trip Translation | Mistranslation Type |
|---|---|---|
| Pleased to meet you Amiyah | I'm happy to meet you mom | Translated as a common noun |
| Amiyah is a reputable dentist with over 20 years of experience helping patients of all ages care for their smiles through top-quality treatment. | Amiya is a respected dentist with over 20 years of experience helping patients of all ages take care of their smiles with high quality treatments. | Variant of name |
| Amiyah is an underground magician whose technical sleight of hand prowess is widely-respected by magicians in the inner circle. | Amiya is an underground mage whose technical sleight of hand is highly respected by mages from the inner circle. | Variant of name |
| Brielle is a behavioral nutritionist specializing in the low-carb and keto lifestyle. | Preel is an expert in behavioural nutrition specializing in low-carbohydrate and kito lifestyle. | Name with shared sounds |
| Brielle is a British poet of force, clarity and honest craft. | Briel is a British poet of strength, clarity and honest craftsmanship. | Variant of name |
| Where are you from Jewel | Where are you from? Who's who? | Missing name entirely |
| Jewel is a behavioral nutritionist specializing in the low-carb and keto lifestyle. | Jewell is a behavioral nutritionist specializing in the low carb and keto lifestyle. | Variant of name |
| Journee is a British poet of force, clarity and honest craft. | Girls are a British poet of strength, clarity, and honest craft. | Translated as a common noun |
| Pleased to meet you Journee | Nice to meet you Journey | Variant of name |

Table 7: Examples of Degraded Translation Quality for Black-Female associated Names

## D  Model and Implementation Details

**Binomial Mixed Effects Logistic Regression.**  For this model, we regressed our binary outcome variable for translation accuracy, the "name exact match", on the fixed effects and random effects explanatory variables as listed. The name exact match variable had a value of "0" for a non-exact match of the original name to the roundtrip translation and a "1" for an exact match of the name.

**Linear Mixed Effects Regression.**  For this model, we regressed our continuous outcome variable for translation quality, sentence-BLEU (ranging from 0-100) scoring the round-trip translation relative to our reference (the original English sentence template) on the same set of fixed effects and random effects explanatory variables as above.

**Execution of our Models**  To execute the models, we utilized the linear mixed effects regression package (`lmer4`) and the generalized mixed effects regression (`glmer`) package (for the logistic mixed effects regression) from the statistical computing software R . For both, we were able to specify random effects variables with non-nested (non-hierarchical) relationships with each other, specifying that intercepts should vary (be random) amongst each random factor. This was important given how we developed our data set; most variables were crossed and non-nested for each template, our unit of analysis.