# OpenReview forum: "A Rose by Any Other Name would not Smell as Sweet: Social Bias in Names Mistranslation"
_EMNLP/2023/Conference — EMNLP 2023 Main_

### Official Review · Reviewer_ndtx · 2023-07-26

**Soundness:** 4

**Excitement:**

4: Strong: This paper deepens the understanding of some phenomenon or lowers the barriers to an existing research direction.

**Paper Topic And Main Contributions:**

The authors measure the difference in how often different names are mistranslated. They use round-trip machine translation for 16 template sentences filled in with about a thousand names (categorized by gender and race/ethnicity). They find that female-associated names and names associated with racial and ethnic minorities are mistranslated more often.

**Reasons To Accept:**

If mistranslation of names has not been studied yet, then this paper brings up an important new type of social bias encoded in language models.

**Reasons To Reject:**

The 16 template sentences were chosen with reasons, but it is possible that these sentences do not completely represent the wide variety of sentences with names in them which appear in "real world" contexts. The authors also discuss the US-centric dataset used in their Limitations section.

**Reproducibility:**

5: Could easily reproduce the results.

**Reviewer Confidence:**

4: Quite sure. I tried to check the important points carefully. It's unlikely, though conceivable, that I missed something that should affect my ratings.

---

> ### Author Rebuttal · Authors · 2023-08-28
>
> Dear Reviewer, Area Chairs, Senior Area Chairs and Program Chairs,
>
> Thank you for the careful review and constructive feedback. We address the main comment and question below.
>
> Review 3: Reviewer ndtx
>
> Reviewer Comment: The 16 template sentences were chosen with reasons, but it is possible that these sentences do not completely represent the wide variety of sentences with names in them that appear in "real world" contexts. The authors also discuss the US-centric dataset used in their Limitations section.
>
> Our Response:
> Thank you for the observation; yes the templates do not fully represent the full variety of sentences that appear in real-world contexts. This is a standard limitation of template-based datasets. We chose our sentence templates in light of our harms, and specifically to align to scenarios where names matter and to be independent from the demographic variables considered, including ensuring gender neutrality of the sentences. We will add a note about this in the Limitations section in a future version of the paper.

---

### Official Review · Reviewer_EGZQ · 2023-08-04

**Soundness:** 2

**Excitement:**

3: Ambivalent: It has merits (e.g., it reports state-of-the-art results, the idea is nice), but there are key weaknesses (e.g., it describes incremental work), and it can significantly benefit from another round of revision. However, I won't object to accepting it if my co-reviewers champion it.

**Paper Topic And Main Contributions:**

This paper investigates the idea that translation quality for names and surrounding context will be lower for names associated with US racial and ethnic minorities. They develop a dataset of demographically aligned names and propose a translation evaluation procedure based on round-trip translation. The study finds that translation systems are significantly less capable of correctly translating female-associated names, especially those associated with racial (Black) and ethnic (Hispanic) minorities.

**Reasons To Accept:**

Shows biases along race and gender lines in machine translation, where there are more errors for some disadvantaged groups when you interact multiple membership together.



**Reasons To Reject:**

The paper says that these mistranslations are due to "the systems' tendencies to standardize language to predominant language patterns." This is a very strong causal interpretation of the results. Is that really justified by the evidence presented? It could be that these mistranslations are due to relative frequencies of these patterns in the MT system training data.

The paper also says these mistranslations have "significant implications for people’s professional, personal and cultural identities, self-worth and ease of communication." That is another very strong causal interpretation of the results. Is that really justified by the evidence presented?

The paper has an unjustified focus on US gender and racial issues in the current time period. Machine translation systems are about languages, not about specific countries at specific time periods.

The abstract seems to oversell the results. There are no differences in translation accuracy for female, black, AAPI, or female AAPI names for one of the outcome measures. They have to get into a few of the specific interactions to find significant effects. This is not explained.

Logistic regression has strong statistical assumptions for inference that are not discussed. Construction of standard errors are not discussed.

**Reproducibility:**

3: Could reproduce the results with some difficulty. The settings of parameters are underspecified or subjectively determined; the training/evaluation data are not widely available.

**Reviewer Confidence:**

3: Pretty sure, but there's a chance I missed something. Although I have a good feel for this area in general, I did not carefully check the paper's details, e.g., the math, experimental design, or novelty.

---

> ### Author Rebuttal · Authors · 2023-08-28
>
> Dear Reviewer, Area Chairs, Senior Area Chairs and Program Chairs,
>
> Thank you for the careful review and constructive feedback. We address the main comments and questions below.
>
> Review 2: Reviewer EGZQ (your comments below followed by our responses)
>
> Comment 1: The paper says that these mistranslations are due to "the systems' tendencies to standardize language to predominant language patterns." This is a very strong causal interpretation of the results. Is that really justified by the evidence presented? It could be that these mistranslations are due to the relative frequencies of these patterns in the MT system training data.
>
> Response to Comment 1: Thank you for describing your interpretation of our statement. Our reading of your description of the likely reason for the translation errors is actually quite similar to the intended meaning of our statement, but we understand that you feel it is strongly worded. As background, subword segmentation such as Byte-Pair Encoding will produce representations or embeddings of the most frequently found patterns in the parallel corpora used to train neural MT systems. To clarify our statement, by “standardize language” we mean that name-related subword representations from the English training corpora are more likely to be from the majority social groups represented, so the roundtrip translations may have a tendency to output these more common names erroneously in cases where the original names were associated with less-represented social groups in the training data. As stated by (Weidinger et al., 2021) “...training data can be biased because some communities are better represented in the training data than others. As a result, LMs (language models) trained on such data often model speech that fails to represent the language of those who are marginalized, excluded, or less often recorded”.
>
> Comment 2: The paper also says these mistranslations have "significant implications for people’s professional, personal and cultural identities, self-worth and ease of communication." That is another very strong causal interpretation of the results. Is that really justified by the evidence presented?
>
> Response to Comment 2:
> To clarify, we do not argue that our Results section nor Conclusion show that these harms are directly having these effects on a specific group of technology users; in order to conclusively show that, we would need to conduct a before-and-after longitudinal study following a representative sample of people from different social groups over time to understand the impacts of their experiences with MT system name translations.
> In the particular statement you reference from our abstract, we are instead suggesting, given our results, that name mistranslations could have long-term adverse impacts in the form of harms as outlined in Section 2 Socio(linguistic) Background. To recap, in the Results section, we report lower translation accuracy and translation quality scores for names associated with females particularly Black and Hispanic females. Given these results and in light of historical evidence that language (including MT) technology has exacerbated harms for vulnerable populations (Weidinger, et al., (2021), Bender, et al., (2021), Blodgett et al., (2020), Savoldi, et al., (2021)), we suggest that lower translation accuracy and quality for vulnerable social groups risk their experiencing allocative harms (Crawford, 2017; Barocas et al., 2017), such as poorer quality of MT system service or poorer professional communications when MT systems incorrectly translate their names (or sentences including their names), or representational harms (Katzman et al., 2023) such as erasure of the name and its associated culture.
>
> Comment 3: The paper has an unjustified focus on US gender and racial issues in the current time period. Machine translation systems are about languages, not about specific countries at specific time periods.
>
> Response to Comment 3:
> We appreciate your alternative perspective and concern. Now that MT systems are routinely used in the real world, they inevitably impact people in many ways (both positive and negative). In line with a growing body of work on understanding and measuring these impacts (Weidinger, et al., (2021), Bender, et al., (2021), Blodgett et al., (2020), Savoldi, et al., (2021)), our study needs to be situated in specific real-world data. To investigate fairness and inclusivity of name translation, where names are a meaningful aspect of languages, we use a dataset of names from the US (and the associated social group distinctions) because it lets us use English as a source language and is publicly available. As discussed in our Limitations and Ethics sections, we do not claim that our results generalize beyond the settings tested.
>
> Comment 4:
> The abstract seems to oversell the results. There are no differences in translation accuracy for female, black, AAPI, or female AAPI names for one of the outcome measures. They have to get into a few of the specific interactions to find significant effects. This is not explained.
>
> Response to Comment 4: This statement does not accurately represent our results. We did find significant differences in translation accuracy for Female, Black, AAPI, and Female AAPI-associated names – please reference Section 6.1 “Correctness of Translation” (these results can be seen in Tables 4 and 5 of our paper; significant effects are denoted with an asterisk after each relevant p-value). We describe in Section 6.1 how translation accuracy results are lower across both female-associated names as a whole, as well as the intersectional combinations (e.g. female-AAPI-associated names) across the board (see odds ratios for differences for each social group in comparison to the baseline of an odds ratio = 1). Name translation quality results as measured by sentence-BLEU can be seen in Section 6.3 Table 6. Our results for translation quality reinforce our finding that Black and Hispanic-female-associated names have lower translation accuracy; specifically, we observe degraded translation quality for sentences including these names. We observed that the translation quality effect sizes have smaller effect sizes than the translation accuracy, but this was expected since BLEU is an average over many words, most of which will probably be correctly translated.
>
> Comment 5:
> Logistic regression has strong statistical assumptions for inference that are not discussed. Construction of standard errors are not discussed.
>
> Response to Comment 5:
> We utilized a logistic mixed effects model which is distinct from a standard logistic regression therefore we chose to express the statistical assumptions related to mixed effects models in Appendix B; these apply to both our logistic and linear mixed effects models. For more information on the logistic mixed effects model regression and the binary outcome variable we used for it (name accuracy where an exact match of the name existed between the roundtrip translation and the original name), please see Section 5.2 specifically “Name Translation Accuracy”, our overall model specification detailed in Section 5.3 and Appendix C (predictors were the same for both logistic and linear mixed effects models, only the outcome variables differed where the one for logistic regression was a binary value of 1 for match and 0 for non-match), and finally the model and implementation details further listed in “BinomialMixed Effects Logistic Regression” under Appendix D. With respect to standard errors, we chose not to include these since we were most interested in determining whether the odds ratios (and corresponding coefficients) for the logistic mixed effects regression were significant and by what degree for a given fixed effect factor (for example, gender*race_ethnicity=AAPI). We would be happy to provide standard errors upon request for the fixed effects factors but these may not be easily interpretable given that all of our predictors are binary variables.
>
> Citations
>
> Solon Barocas, Kate Crawford, Aaron Shapiro, and Hanna Wallach. 2017. The problem with bias: Allocative versus representational harms in machine learning. In Proceedings of SIGCIS, Philadelphia, PA. The Special Interest Group for Computing, Information and Society.
>
> Emily M. Bender, Timnit Gebru, Angelina McMillan-Major, and Shmargaret Shmitchell. 2021. On the Dangers of Stochastic Parrots: Can Language Models Be Too Big? In Proceedings of the 2021 ACM Conference on Fairness, Accountability, and Transparency (FAccT '21). Association for Computing Machinery, New York, NY, USA, 610–623. https://doi.org/10.1145/3442188.3445922
>
> Su Lin Blodgett, Solon Barocas, Hal Daumé III, and Hanna Wallach. 2020. Language (Technology is Power: A Critical Survey of “Bias” in NLP. In Proceedings of the 58th Annual Meeting of the Association for Computational Linguistics, pages 5454–5476, Online. Association for Computational Linguistics.
>
> Kate Crawford. 2017. The trouble with bias. Conference on Neural Information Processing Systems (NeurIPS).
> Beatrice Savoldi, Marco Gaido, Luisa Bentivogli, Mateo Negri, and Marco Turchi. 2021. Gender bias in machine translation. Transactions of the Association for Computational Linguistics, 9:845–874.
>
> Weidinger, L., Mellor, J., Rauh, M., Griffin, C., Uesato, J., Huang, P. S., ... & Gabriel, I. (2021). Ethical and social risks of harm from language models. arXiv preprint arXiv:2112.04359.

---

### Official Review · Reviewer_QCvK · 2023-08-04

**Soundness:** 4

**Excitement:**

3: Ambivalent: It has merits (e.g., it reports state-of-the-art results, the idea is nice), but there are key weaknesses (e.g., it describes incremental work), and it can significantly benefit from another round of revision. However, I won't object to accepting it if my co-reviewers champion it.

**Missing References:**

- Hermjakob, Ulf, Kevin Knight, and Hal Daumé III. "Name translation in statistical machine translation-learning when to transliterate." *Proceedings of ACL-08: HLT*. 2008.
- Maurel, Denis, et al. "Prolex: a lexical model for translation of proper names. Application to French, Serbian and Bulgarian." *Bulag-Bulletin de Linguistique Appliquée et Générale* (2007): 55-72.

**Paper Topic And Main Contributions:**

This paper carries out an in-depth analysis of the biases in translation quality of contexts and names associated with different gender and racial/ethnic groups. Specifically, the authors use a dataset of baby names and associated demographic data, and embed each name into different sentence context templates. They then quantify the disparity of error rates in round-trip (English → Intermediate Language → English) translations of each sentence while controlling for intermediate language, translation method, and sentence template. They provide quantitate (mixed-effect model coefficient) and qualitative error analysis of error rates in absolute match and BLEU scores associated with name gender, racial/ethnic groups.

**Contributions:**

- A dataset for testing demographic biases in machine translation named DNIC (Demographic Names in Context).
- A mixed-effects model setup for quantifying the demographic biases across MT systems.

**Questions For The Authors:**

- Question A: Are name variant errors equivalent to translation errors?
- Question B: How are the race categories determined in the NYC baby names data? Do certain categories have more internal diversity?

**Reasons To Accept:**

- Good statistical modeling and analysis: This quantitative methodology for mixed effects modeling, both classification and regression, provides a good example of measurement of bias across related conditions/system for MT models .
- New dataset for names and contexts to measure MT system biases: The dataset is generated from a New York City baby names dataset, with demographics derived from log-odds associated with the conditional likelihood of a name belonging to a group. Sentence templates are selected from scenarios that are unlikely to have demographic dependence (interaction and biographies) making this a well-designed dataset.

**Reasons To Reject:**

- Issues with error measurement: The qualitative analysis of errors (sec 6.3) indicates that a majority of errors are due to a variant/spelling of the same name (eg. Hanna → Hannah). It’s unclear whether these should qualify as an error since this may be an artifact of different phonemic natures of the orthography of English and the intermediate language. The authors should consider using non-exact match errors (eg. phonetic/edit distance based) as the focus is on the name translation errors.
- Non-uniqueness of round trip mapping: Related to the error measurement, the authors claim that name translation quality cannot be directly evaluated by checking source→target MT as target languages may use alternative acceptance criteria for name translations. It is unclear why this same critique does not extend to the round trip task that is the focus of the paper. The authors may consider reducing the variety of target languages and consider the delving deeper into language specific conventions for translations of proper names (eg Hermjakob et al. (2008), Maurel et al. (2007)).

**Reproducibility:**

5: Could easily reproduce the results.

**Reviewer Confidence:**

4: Quite sure. I tried to check the important points carefully. It's unlikely, though conceivable, that I missed something that should affect my ratings.

---

> ### Author Rebuttal · Authors · 2023-08-28
>
> Dear Reviewer, Area Chairs, Senior Area Chairs and Program Chairs,
>
> Thank you for the careful review and constructive feedback. We address the main comments and questions below.
>
> Reviewer QCvK (your paraphrased comments below followed by our responses):
>
> Comment 1 (“Issues with error measurement”):
> Reviewer noted that most translation errors were variants of the same name and wondered (since they are potentially linguistic artifacts) if name variants should be considered errors (reviewer question A); as such, the reviewer also recommended considering non-exact match errors (eg. phonetic/edit distance based).
>
> Response to Comment 1 and Question A: Our position is that name variant errors are equivalent to translation errors. We chose to evaluate errors relative to the exact translation of United States (US)-assigned baby names from birth records since exact spellings are frequently associated with cultural identities of racial/ethnic subgroups and possibly personal preference. We acknowledge that different people may accept different spelling variations but we have no way of knowing these differing preferences or the desired translation system behaviors without conducting a human study to gather these preferences. Your questions and the suggestion to include non-name exact matches are helpful; we agree that such a measure will be a useful comparison for our later version of the paper.
>
> Comment 2(“Non-uniqueness of roundtrip mapping”): The reviewer basically asked why round trip machine translation (MT) is suited to our evaluation versus translation of source to target language questioning specifically why the use of alternative acceptance criteria for name translations in target languages would not also pose an issue with round trip MT. They also suggested we narrow to a couple of languages to explore those languages’ conventions for first-name translations, pointing us to additional references.
>
> Response 2: We utilized a round-trip evaluation approach because it allowed us to provide the closest approximation of an error rate possible for name translations using automatic translation, i.e. MT. Manual evaluation of name translations in each direction would require a significant amount of expertise and labor. While we note (see Section 5.2 Round Trip Translation) that it is difficult to pinpoint where any translation errors may occur with this approach (in the forward pass or the backward pass of the round trip translation), since we use a mixed effects regression approach to assess translation accuracy and translation quality, we are able to focus primarily on gender and race/ethnicity (our fixed effects) on name translation errors. In particular, we control for the error contributed by template, pivot language, and MT system, so that we can isolate the fixed effects and their intersections (interactions) on name translation. However, your suggestion to explore conventions for name translations in other languages (which potentially impact the overall roundtrip translation error rate) might shed light on the causes of the errors, and thus assist with future mitigation of name mistranslations. We thank you for the additional suggestion and references.
>
> Reviewer Question B: How are the race categories determined in the NYC baby names data? Do certain categories have more internal diversity?
>
> Response to Reviewer Question B: Race/ethnicity for a baby name is based on that documented for the birth parent from NYC governmental birth records. We are not clear on what specific information you seek in the second question, but please reference Section 4.1 Table 2 in case you are looking for the distribution (count) of names by race/ethnicity and gender categories.
>
> Missing References: We will include the recommended references. Thank you!

---

### Meta-Review · Area_Chair_v3Ui · 2023-09-20

**Recommendation:** 4

**Metareview:**

This paper presents an analysis with the objective of comprehending biases within machine translation systems concerning names across various dimensions, including gender and racial/ethnic groups. The authors have also contributed a dataset to facilitate the examination of this issue. The paper offers an intriguing perspective on biases in machine translation systems, supported by a comprehensive experimental framework.

The authors have provided robust responses to the reviewers' critiques, particularly in addressing the concerns raised by Reviewer 2. I concur with Reviewer 1's point that name variant errors should not be equated with translation errors, even though both aspects are crucial to assess. I strongly encourage the authors to incorporate this suggestion into the paper, as articulated in their response to the reviewers.

In summary, this work contributes to our understanding of biases in machine translation systems. By addressing the reviewers' feedback, particularly the suggestion from Reviewer 1, the paper can further enhance its clarity and impact.

---

### Decision · Program_Chairs · 2023-10-07

**Decision:**

Accept-Main

**Comment:**

This paper presents an analysis with the objective of comprehending biases within machine translation systems concerning names across various dimensions, including gender and racial/ethnic groups. The authors have also contributed a dataset to facilitate the examination of this issue. The paper offers an intriguing perspective on biases in machine translation systems, supported by a comprehensive experimental framework.

The authors have provided robust responses to the reviewers' critiques, particularly in addressing the concerns raised by Reviewer 2. I concur with Reviewer 1's point that name variant errors should not be equated with translation errors, even though both aspects are crucial to assess. I strongly encourage the authors to incorporate this suggestion into the paper, as articulated in their response to the reviewers.

In summary, this work contributes to our understanding of biases in machine translation systems. By addressing the reviewers' feedback, particularly the suggestion from Reviewer 1, the paper can further enhance its clarity and impact.